# Characterization of RNA Helicase Genes in *Ustilago maydis* Reveals Links to Stress Response and Teliospore Dormancy

**DOI:** 10.3390/ijms26062432

**Published:** 2025-03-08

**Authors:** Amanda M. Seto, Barry J. Saville

**Affiliations:** 1Environmental & Life Sciences Graduate Program, Trent University, Peterborough, ON K9L 0G2, Canada; amandaseto@trentu.ca; 2Department of Forensic Science, Trent University, Peterborough, ON K9L 0G2, Canada

**Keywords:** teliospore dormancy, germination, RNA helicase, stress response, *uded1*, *udbp3*, *Ustilago maydis*

## Abstract

Fungi produce dormant structures that are responsible for protection during adverse environmental conditions and dispersal (disease spread). *Ustilago maydis*, a basidiomycete plant pathogen, is a model for understanding the molecular mechanisms of teliospore dormancy and germination. Dormant teliospores store components required for germination including mRNAs which may be stored as dsRNAs. RNA helicases are conserved enzymes that function to modulate, bind, and unwind RNA duplexes, and can displace other proteins. We hypothesize that RNA helicases function during teliospore dormancy to stabilize and/or modulate stored mRNAs. We identified the *U. maydis udbp3* and *uded1* as encoding RNA helicases of interest as they are upregulated in the dormant teliospore and decrease during germination. Experimental results suggest that *udbp3* may function as a negative regulator of osmotic stress-responsive genes and that *uded1* modulates stress response by repressing translation. The altered expression of *uded1* also results in slow growth, polarized growth, and the formation of dsRNA. Together, the data support a role for both helicases modulating gene expression, in response to stress, leading to teliospore dormancy and also modulating responses for teliospore germination. Increasing our molecular understanding of these processes will aid in developing novel strategies to mitigate disease spread.

## 1. Introduction

Dormancy is defined as a time of rest or pause in phenotypic development [1]. Many organisms develop dormant structures as a strategy to survive adverse environmental conditions. For example, some plants develop dormant seeds to withstand unfavourable environmental conditions until favourable conditions are met for germination to occur [2]. Some microbes utilize dormancy as a response to environmental stress. This allows for the microbe to maintain their viability until favourable growth conditions are met [3]. Fungi develop spores that enable their survival for long periods of time and their dispersal allows for disease spread. Characteristic traits of dormant fungal spores are limited cell proliferation and low metabolic activity. Once conditions are favourable, there is an irreversible transition to germination where respiration and metabolic rates increase [1]. Understanding the molecular mechanisms that occur during this transition from dormancy to germination can aid in developing methods for mitigating disease spread among fungal plant pathogens. We use the basidiomycete *Ustilago maydis* (DC.) Corda as a model for studying the teliospore transition from a dormant state to an actively germinating spore.

*Ustilago maydis* is a plant pathogen that infects *Zea mays* resulting in the development of the disease Common Smut of Corn. The disease is characterized by the growth of tumours that contain billions of black teliospores. These tumours crack open, releasing teliospores into the environment, enabling the spread of disease to other crops. The dormant teliospores contain three-layer-thick cell walls that are melanized and echinulated [4,5,6]. Teliospores can remain viable for years [7] and germination is stimulated in the presence of a carbon source [8]. Germination is the irreversible transition from low to high metabolic activity where the spore utilizes its stored reserves to facilitate the resumption of metabolism. Phenotypic development also resumes and a germ tube is produced [9]. Fungal spores contain the necessary components that are required for the initiation of germination. These components can include carbohydrates as energy reserves and stored RNAs [10]. Previous research indicates that dormant *U. maydis* teliospores contain stored RNAs bound to proteins to form a complex. These ribonucleoprotein (RNP) complexes disappear following germination and may function to protect the RNA during dormancy [11,12]. Our laboratory hypothesized that dormant *U. maydis* teliospores contain mRNAs that are stabilized by binding with complementary antisense RNAs to form double-stranded RNA (dsRNA) [13,14]. We hypothesize that RNA helicases are involved in stabilizing RNA:RNA interactions and unwinding mRNA transcripts for translation following the initiation of germination.

RNA helicases are highly conserved enzymes that function in all aspects of RNA metabolism. They are capable of driving and regulating gene expression by remodelling RNA, unwinding RNA duplexes, displacing proteins, and binding to RNA to create ribonucleoprotein (RNP) complexes [15,16,17]. Classification of RNA helicases is based on their sequence and structure motifs as described by Gorbalenya and Koonin [18] and further analyzed by Singleton et al. [19]. A total of six helicase superfamilies have been identified with superfamilies 1 (SF1) and 2 (SF2) being the largest. The majority of eukaryotic RNA helicases can be found in SF2 [19,20].

The functional characterization of RNA helicases in several eukaryotes has revealed that these enzymes have roles in cellular and metabolic pathways. For example, the RNA helicase *VAD1* in *Cryptococcus neoformans* is associated with the regulation of several virulence genes, response to stress, and salt tolerance [21]. Deletion of the *Magnaporthe oryzae* RNA helicase *MoDHX35* results in reduced appressoria formation and attenuated virulence [22]. Despite the vast research on the function of RNA helicases, their roles in the lifecycle of phytopathogenic fungi has not been extensively explored.

RNA-seq analysis identified several RNA helicases that are upregulated in the dormant teliospore and decrease during germination [23]. Specifically, pattern 17 contained transcripts that were upregulated in the dormant teliospore, decreased during germination, and remained decreased as germination progressed. Of the five RNA helicases identified in this pattern, the orthologs to *Saccharomyces cerevisiae DBP3* and *DED1* were selected for further characterization in *U. maydis*. In budding yeast, *DBP3* is an SF2 DEAD-box RNA helicase that is not essential for cell viability and is involved in processing the ITS1 A3 cleavage site during pre-RNA maturation [24]. Delaney et al. [25] demonstrated that *DBP3* deletion mutants had increased thermotolerance and were resistant to oxidative, endoplasmic reticulum, and DNA damage stressors. This suggests that the *U. maydis* ortholog, *udbp3,* may also function in stress response. The *DED1* RNA helicase is essential with roles in translation promotion and repression [26,27,28]. We hypothesized that the *U. maydis* ortholog, *uded1*, is involved in repressing translation during teliospore dormancy and modulating translation when germination is initiated. The results of this study suggest that *udbp3* is a negative regulator of osmotic stress response. Upregulation of *udbp3* during dormancy may function to regulate a subset of stress-responsive genes. In contrast, *uded1* may modulate the translation of genes during dormancy and germination. This study identifies potential molecular mechanisms that are involved during teliospore dormancy and germination and offers insight into gene regulation during the transition from dormancy to germination.

## 2. Results

### 2.1. Identification of RNA Helicases with Potential Roles During Teliospore Germination

The transcriptome data from Seto, Donaldson and Saville [23] identified five candidate RNA helicases that we hypothesized to have a role during *U. maydis* teliospore germination. These RNA helicases were found in pattern 17 of teliospore germination and have transcript levels that were decreased in the haploid and dikaryon cell types, upregulated in the dormant teliospore, and decreased during teliospore germination. This specific transcript pattern suggested a role during the exit from dormancy to germination. Candidate RNA helicases were identified as *UMAG_04080*, *UMAG_01732*, *UMAG_10241*, *UMAG_01122*, and *UMAG_00835*. Seto and Saville [29] identified these RNA helicases as orthologs to *S. cerevisiae DED1*, *DBP3*, *DBP8* and *HCS1,* respectively. *UMAG_00835* is a putative basidiomycete-specific RNA helicase with possible functions in ribosome biogenesis [29]. The RNA helicases *UMAG_01732* and *UMAG_04080* were prioritized as genes of interest based on their confirmed transcript level pattern and the observations that their transcript levels were the highest in the dormant teliospore compared to the other candidates [23].

Bioinformatic analysis of *UMAG_01732* was previously performed by Seto and Saville [29] and was identified as a putative fungal- and plant-specific RNA helicase. UMAG_01732 contains a helicase core consistent with the sequence motifs previously identified in *S. cerevisiae* Dbp3 (Figure 1A). Weaver, Sun and Chang [24] previously identified the signalling sequence motif lysine-lysine-X (KKX) repeated several times in *S. cerevisiae* Dbp3. Our results of the protein MUSCLE alignment with putative Dbp3 orthologs indicated conservation of the KKX sequence motif; however, the repeat length of the motif differs between species. The *U. maydis* KKX motif is repeated five times compared to the 10 tandem repeats in *S. cerevisiae*. A phylogenetic tree was created from the MUSCLE alignment of known and putative Dbp3 orthologs and the closely related Dbp2/DDX17 RNA helicases. The phylogenetic tree showed two distinct clades from the SF2 DEAD-box RNA helicases (Figure 1B). In Figure 1B, the boxed region indicates the Dbp3 group of RNA helicases that contains UMAG_01732. These results are consistent with our previous phylogenetic trees that identified UMAG_01732 as a putative fungal and plant RNA helicase [29] and we have named the *U. maydis* ortholog as Udbp3.

The *S. cerevisiae* Ded1 is a well-characterized SF2 DEAD-box RNA helicase and the *H. sapiens* ortholog was identified as DDX3. Protein sequence and phylogenetic analysis conducted by Seto and Saville [29] identified UMAG_04080 as the putative ortholog. UMAG_04080 contains a helicase core consistent with known Ded1/DDX3 proteins in other organisms (Figure 2A). A phylogenetic tree was created with Ded1/DDX3 orthologs and the closely related Tif1/DDX2 orthologs (Figure 2B). The phylogenetic tree shows two distinct clades (Figure 2B), and the boxed region indicates the Ded1/DDX3 clade containing UMAG_04080. These results were consistent with previous results in Seto and Saville [29] and supported UMAG_04080 as an ortholog to Ded1/DDX3. We have named the *U. maydis* ortholog Uded1.

### 2.2. udbp3 Characterization

The ability of Δ*udbp3* mutants to form dikaryotic filaments was assessed through a mating assay. Sexually compatible haploid strains (518 and 521) were combined and spotted on PDA plates containing charcoal. A Fuz^+^ phenotype indicates successful fusion of haploid strains. Filamentous growth was unaffected in the reciprocal (Δ*udbp3* × wt) or deletion (Δ*udbp3* × Δ*udbp3*) crosses. The deletion of *udbp3* did not affect mating ability (Appendix A).

A pathogenesis assay was performed to determine the ability of *udbp3* mutants to infect the plant. Golden Bantam *Z. mays* seedlings were infected with the reciprocal, deletion, or wild-type crosses. Both reciprocal and deletion strains showed no significant difference in virulence compared to wild-type infections (Appendix A). The infected seedlings were able to produce tumours and develop teliospores. This indicated that the deletion of *udbp3* did not affect infection, disease development, tumour formation, or teliospore development.

Teliospores were obtained from Golden Bantam *Z. mays* ears infected with the deletion (Δ*udbp3* × Δ*udbp3*) strains and germination was assessed through teliospore germination time courses. Germination was assessed at 16 h post induction of germination (PIG), germination percentage was determined, and morphology was examined with microscopy. There was no visual difference in the morphology of dormant teliospores. Teliospore germination was not affected, no visible defects of the promycelium were detected, and basidiospores were still produced suggesting that meiosis was not affected (Appendix A). The combined results indicated that *udbp3* was not required for the progression of teliospore germination.

*Saccharomyces cerevisiae DBP3* deletion mutants have a slow-growth phenotype [24], increased thermotolerance, and were tolerant to endoplasmic reticulum (ER) stress caused by tunicamycin, oxidative stress caused by paraquat, and resistant to DNA damage caused by exposure to methyl methane sulfonate (MMS) [25]. *Ustilago maydis* Δ*udbp3* mutants were assessed for abnormalities in growth and stress response. We found no difference in the growth of the 518Δ*udbp3* and the 521Δ*udbp3* strains when compared to the wild-type when incubated at 16 °C, 28 °C, and 37 °C (Appendix A). The Δ*udbp3* mutants also showed no difference, compared to wild-type strains, in growth when exposed to tunicamycin, paraquat, or MMS (Appendix A).

The response to osmotic stress was not previously assessed in *S. cerevisiae DBP3* mutants. The response to osmotic stress was assessed in the Δ*udbp3* mutants by spotting a 10-fold dilution series on YEPS plates containing 1 M NaCl. Of the 518Δ*udbp3* mutants, two of the biological replicates showed no difference in growth compared to 518. The 518Δ*udbp3* #1 mutant showed a slight decrease in growth compared to the other deletion biological replicates and 518 (Figure 3). The 521Δ*udbp3* mutants showed an increased tolerance to osmotic stress compared to 521 (Figure 3). The parental stains 518 and 521 are sister strains [33] and have previously described differences in growth and hormone concentrations, so the differences in osmotic stress response were not unexpected [34].

### 2.3. Deletion of uded1 Is Detrimental to Growth

Ded1 in *S. cerevisiae* is an essential protein for cell viability. To assess the importance of *uded1* in *U. maydis*, we attempted to create *uded1* deletion mutants using the Kämper [35] homologous recombination-based method. These attempts resulted in no viable mutants. A two-step gene deletion method described by Ostrowski and Saville [14] was used to create *uded1* deletion mutants. This method involved first creating expression mutants where the ectopic expression of *uded1* was placed under the control of a carbon-sensitive inducible promoter and integrated at the *ip* locus. These expression mutants (crg1:*uded1*) were created in the sexually compatible 518 and 521 wild-type strains so that we could later assess the impact of gene alterations on the ability of the fungus to mate and infect *Z. mays*. Homologous recombination was then used to replace the native *uded1* with a hygromycin B resistance cassette in the crg1:*uded1* strains. Growth of deletion strains (Δ*uded1* crg1:*uded1*) was observed when incubated in the presence of L-arabinose (permissive growth conditions). In repressive growth conditions, where D-glucose is the carbon source, these mutants were effectively deletion mutants (Figure 4A). Under repressive growth conditions, the deletion mutants showed reduced growth at the 10^−1^ and 10^−2^ dilutions compared to the wild-type and expression strains. No difference in growth was found between the 518Δ*uded1* crg1:*uded1* and 521Δ*uded1* crg1:*uded1* strains (Appendix A). The ability to control when the ectopically integrated *uded1* is expressed thus enabled us to determine that *uded1* is required for full cell viability and growth in *U. maydis*.

Another growth phenotype that was observed in the Δ*uded1* crg1:*uded1* mutants was the length of time mutants took to grow on solid medium in permissive conditions compared to the wild-type and expression strains. Following incubation at 28 °C for 7 days, the wild-type and crg1:*uded1* strains had significantly more growth than the Δ*uded1* crg1:*uded1* strains. The growth of the Δ*uded1* crg1:*uded1* mutant colonies was significantly smaller compared to the wild-type and expression strains (Figure 4B).

During our early experiments with the Δ*uded1* crg1:*uded1* mutants, a mycelial growth phenotype was observed when these strains were streaked on YEPA plates supplemented with hygromycin B. This phenotype was also observed on YEPA, DCMA, and CMA plates supplemented with and without hygromycin B. In *U.*
*maydis*, the switch from budding haploid cells to filamentous growth is in response to environmental factors. Changes in nutrient availability [36], exposure to air [37], acidic pH [38], triacylglycerides [39], and involvement of the cyclic AMP/protein kinase A pathway [37,40]. In order to assess the cell morphology of the Δ*uded1* crg1:*uded1* mutants, all strains were initially streaked on DCMA plates containing 1 M sorbitol. Single colonies were picked and patched onto either YEPA or YEPA containing 1 M sorbitol plates. Following a 3-day incubation at 28 °C, the 518Δ*uded1* crg1:*uded1* patches grown on YEPA appeared slightly fuzzy compared to both wild-type and 518 crg1:*uded1* strains (Figure 5). The 518Δ*uded1* crg1:*uded1* strains grown on YEPA with 1 M sorbitol did not differ in appearance compared to wild-type strains. The cell morphology of all strains was examined through microscopic analysis. For 518Δ*uded1* crg1:*uded1* stains grown in YEPA, a mixture of normal budding cells and cells that appear long and filament-like was observed. When the same strains are grown on YEPA with 1 M sorbitol, the cells appear indistinguishable from wild-type (Figure 5). This mycelial growth phenotype was also seen in the 521Δ*uded1* crg1:*uded1* mutants when grown on YEPA and normal budding growth was restored when cells were grown in the presence of sorbitol (Appendix A). This indicates that the osmotic stabilizing affects of sorbitol are required for the mutant strains to grow in a manner similar to wild-type strains.

### 2.4. The Ability for uded1 Mutants to Mate and Infect the Plant

A mating assay was performed to assess if the altered expression of *uded1* impacts the ability of compatible haploid cells to fuse and form a dikaryon. The dikaryon is white and appears fuzzy (Fuz^+^) when spotted on medium containing charcoal. The *uded1* mutant and wild-type strains were cultured in permissive conditions (DCMA) and equal volumes of compatible haploids were mixed and spotted on PDA plates containing charcoal. Plates were incubated at room temperature for 3 days and monitored for Fuz^+^ development. In general, there was no difference in dikaryon formation when wild-type was crossed with the deletion (wt × Δ*uded1* crg1:*uded1*) (Figure 6A). There is a reduction in the density of Fuz^+^ in the wt × crg1:*uded1* crosses where the spots appear less white when compared to the 518 × 521 cross. Fuz^+^ formation is further decreased in the deletion crosses (Δ*uded1* crg1:*uded1* × Δ*uded1* crg1:*uded1*) compared to the 518 × 521 and wt × crg1:*uded1* crosses. The formation of Fuz^+^ was detected when spots were viewed under a stereoscopic microscope (Appendix A). The results of the growth experiments (Figure 5) suggested that if mutants were grown in permissive conditions supplemented with 1 M sorbitol, an osmoprotectant, normal budding growth would result.

A second mating assay (Figure 6B) was conducted where all mutants and wild-type strains were cultured in DCMA containing 1 M sorbitol. The formation of Fuz^+^ on PDA plates containing charcoal was monitored for 3 days. There was no difference in dikaryon formation when expression strains were crossed with wild-type or the deletions (wt × crg1:*uded1* or crg1:*uded1* × Δ*uded1* crg1:*uded1*) and with reciprocal crosses (wt × Δ*uded1* crg1:*uded1*). However, reduced Fuz^+^ development was observed in all deletion crosses (Δ*uded1* crg1:*uded1* × Δ*uded1* crg1:*uded1*) (Figure 6B). The Δ*uded1* crg1:*uded1* × Δ*uded1* crg1:*uded1* spots appeared less dense and were smaller when compared to the 518 × 521 spot. However, the density of the Fuz^+^ formation was greater when compared to the previous mating assay (Figure 6A). Fuz^+^ development in the Δ*uded1* crg1:*uded1* × Δ*uded1* crg1:*uded1* was monitored for up to 5 days at RT but there was no difference in dikaryon formation compared to 3 days (Appendix A). This indicates that with these growth conditions, a wild-type × mutant mating was indistinguishable from wild-type × wild-type but that mutant × mutant matings had substantially reduced Fuz^+^.

The ability for Δ*uded1* crg1:*uded1* mutants to infect the plant was assessed through pathogenesis assays. Equal volumes of compatible haploids were mixed and injected into 7-day-old Golden Bantam seedlings. The Δ*uded1* crg1:*uded1* strains were unable to infect seedlings (Appendix A) and therefore teliospores could not be produced.

### 2.5. Transcript Levels of uded1 in Mutants Grown in Repressive and Permissive Conditions

The transcript level of *uded1* in expression and deletion mutants was assessed when grown in repressive and permissive conditions. In permissive conditions, the *uded1* transcript in the crg1:*uded1* mutants is upregulated compared to the level present in cells grown in repressive conditions and wild-type cells (Figure 7). The *uded1* transcript level in the 518Δ*uded1* crg1:*uded1* mutants was upregulated to an even higher degree than in the 518 crg1:*uded1* mutant (Figure 7A). The same transcript level trend was seen in the 521Δ*uded1* crg1:*uded1* mutants in comparison to the 521 crg1:*uded1* mutant in permissive conditions (Figure 7B). It is the upregulation of *uded1* that is likely contributing to the slow-growth phenotype in Figure 4B, a similar slow growth on over expression was noted for *S. pombe* [26]. In addition, in *S. pombe*, the altered expression of *ded1* contributes to modulating antisense RNAs [41].

### 2.6. Impact of uded1 Expression on dsRNA Stability

Altered *uded1* expression and its impact on sense/antisense interactions was assessed by creating mutants that express *as-ssm1* from an autonomously replicating vector in the crg1:*uded1* mutants. The *as-ssm1* transcript was previously shown to be preferentially expressed in the dormant teliospore and its expression forms dsRNA [14]. This makes it the ideal candidate for assessing sense/antisense interactions when *uded1* expression is altered. The expression of *uded1* was altered by growing the crg1:*uded1* [pCMas-ssm1] mutants in repressive and permissive conditions to repress or activate the crg1 promoter. An S1 nuclease protection assay was performed to determine the presence of dsRNA and assess the impact of *ssm1*/*as-ssm1* interactions when the expression of *uded1* is altered in the crg1:*uded1* [pCMas-ssm1] mutants.

Strand-specific semi-quantitative RT-PCR was used to confirm expression of the *ssm1* and *as-ssm1* transcripts in the crg1:*uded1* [pCMas-ssm1] mutants grown in repressive and permissive growth conditions (Figure 8). The *ssm1* transcript was detected in the empty vector ([pCM768]) control samples and the crg1:*uded1* [pCMas-ssm1] mutants grown in repressive and permissive conditions. The *as-ssm1* transcript was detected in the crg1:*uded1* [pCMas-ssm1] mutants grown in repressive and permissive conditions. Additionally, the *as-ssm1* transcript was detected at low levels in the empty vector controls. The detection of low *as-ssm1* transcript levels was also detected in the haploid cell RNA transcriptome data from Donaldson et al. [42] and Seto, Donaldson and Saville [23]. An internal *UMAG_gapdh*-specific primer was included as a control to assess *UMAG_gapdh* transcript levels. It was noted that in the no first-strand primer (water-primed) samples, an RT-PCR product was produced. This is an indication of non-specific or false-priming during reverse transcription caused by the presence of hairpin structures in the RNA, as previously reported by Ho et al. [43].

The upregulation of *uded1* in the crg1:*uded1* [pCMas-ssm1] mutants was verified using RT-qPCR (Figure 9). All three biological replicates of the 518 crg1:*uded1* [pCMas-ssm1] mutants show an upregulation of *uded1* in permissive conditions compared to growth in repressive conditions (Figure 9A). The same trend was observed in the 521 crg1:*uded1* [pCMas-ssm1] mutants (Figure 9B). One 518 crg1:*uded1* [pCM768] control biological replicate was removed from further analysis as *uded1* was not upregulated in the permissive grown conditions compared to the repressive sample (Appendix A).

The S1 nuclease protection assay was used to determine differences in *ssm1*/*as-ssm1* formation when the expression of *uded1* is altered. All mutant samples expressing *as-ssm1* (crg1:*uded1* [pCMas-ssm1]) from the vector have increased resistance to S1 nuclease digestion in both repressive and permissive conditions indicating that altered expression of *uded1* has no impact on dsRNA formation (Figure 10). The 521 crg1:*uded1* [pCM768] empty vector controls grown in permissive conditions showed increased resistance to S1 nuclease digestion (Figure 10B); however, the same was not observed with the 518 crg1:*uded1* [pCM768] empty vector controls (Figure 10A). The formation of dsRNA in the empty vector controls suggests that upregulation of *uded1* induces conformational changes to the RNA under these conditions or that Uded1 is binding to this region of the transcript to form an mRNP.

## 3. Discussion

The role of RNA helicases in fungal phytopathology is not fully understood and little is known of their roles in fungal virulence and pathogenicity. Our previous work identified 46 RNA helicases in the basidiomycete *U. maydis* [29]. We utilized the RNA-seq data from Seto, Donaldson and Saville [23] to identify candidate RNA helicases that may have a role during *U. maydis* teliospore dormancy and germination. We determined that RNA helicases that are upregulated in the dormant teliospore and then decreased once germination is initiated were RNA helicases of interest. We identified five candidate RNA helicases and focused our investigation on the RNA helicases *udbp3* and *uded1*.

### 3.1. The Role of udbp3 in Osmotic Stress Response

*UMAG_01732* (*udbp3*) was identified as an ortholog of *DBP3* in *S. cerevisiae*. Phylogenetic analysis identified this DEAD-box RNA helicase as fungal- and plant-specific [29]. Udbp3 contains the characteristic nuclear localization signal KKX sequence motif first described by Weaver, Sun and Chang [24] in the *S. cerevisiae* ortholog. This bioinformatic analysis and the transcript pattern of interest during teliospore germination supported the creation of deletion mutants to explore its potential impact on teliospore dormancy and germination.

The deletion of *udbp3* did not adversely affect *U. maydis* virulence, pathogenicity, or teliospore germination. Mutants in *S. cerevisiae* had a slow-growth phenotype at a decreased temperature [24]; however, this was not an obvious phenotype in the *U. maydis* deletion mutants. The *S. cerevisiae DBP3* mutant had been previously characterized for its response to various stressors. The *DBP3* mutant displayed increased tolerance to ER, oxidative, and DNA stress [25]; however, this phenotype was not observed in our *U. maydis* deletion mutants. Our results suggest in the absence of *udbp3*, there is increased tolerance to osmotic stress induced by exposure to 1 M NaCl. *DBP3* orthologs have been identified in other plant species, such as *Arabidopsis thaliana*, *Glycine max*, *Oryza sativa*, *Vitis vinifera*, *Medicago truncatula*, *Malus domestica*, and *Hordeum vulgare*. Characterization of this gene in plants has been limited to *A. thaliana* and *G. max* [44,45]. In *A. thaliana*, the ortholog *strs*1 is a negative regulator of stress-responsive transcription activators and their downstream targets. This RNA helicase is downregulated by saline, osmotic, and heat stress resulting in the enhancement of transcription factors that respond to these stressors. Overexpression of *strs1* leads to decreased heat and saline tolerance [45,46]. In contrast, the *G. max* ortholog, *GmRH*, is induced in low temperatures and during high-salinity conditions. It was proposed that *GmRH* may have a function in processing RNA during low temperature and salt stress conditions [44]. The increased tolerance to osmotic stress in the Δ*udbp3* mutants suggests that in *U. maydis udbp3* may function as a negative regulator in response to stress, similar to *strs1*. The absence of *udbp3* may induce the transcription of stress-related genes and/or their transcription activators.

*udbp3* is upregulated in the dormant teliospore [23] which suggests a function during teliospore dormancy. Based on our current data and previous work in Chung, Cho, Yun, Choi, So, Lee and Lee [44] and Kant, Kant, Gordon, Shaked and Barak [45], *udbp3* may function as a negative regulator of stress-related genes or their activators in response to stress in the teliospore. This would lead to the suppression of stress-related genes in the dormant teliospore. The teliospore comprises physical barriers to protect itself from harsh environmental conditions. The *U. maydis* teliospore cell wall is three layers thick where one layer is melanized and contains ornamentation [6,47,48]. These features allow survival of the teliospore in adverse environmental conditions, such as temperature fluctuations and UV exposure.

### 3.2. The DED1 Ustilago maydis Ortholog Is uded1

Our previous annotation of RNA helicases in *U. maydis* [29] and transcriptomic analyses [23] identified *UMAG_04080* as an RNA helicase of interest. It is an RNA helicase found in pattern 17 of transcript level patterns identified in Seto, Donaldson and Saville [23] where the transcript is upregulated in the dormant teliospore and decreases during germination. The protein sequence analysis and phylogenetic work (Figure 2) identified UMAG_04080 as the ortholog to Ded1 in *S. cerevisiae* and DDX3 in *H. sapiens* [29] and therefore we named this gene *uded1*. The *S. cerevisiae DED1* is an essential gene and deletion mutants have a lethal phenotype [49]. Based on this, it was important to determine the phenotype of the *uded1* mutants. The Δ*uded1* crg1:*uded1* mutants showed that cell growth is inhibited in the absence of *uded1* expression (Figure 4). When these mutants are grown in permissive conditions, growth is noticeably slower than the wild-type and expression strains (Figure 4B). The upregulation of *uded1* in the Δ*uded1* crg1:*uded1* mutants (Figure 7) may contribute to a slow-growth phenotype. The upregulation is caused by the crg1 promoter which yields high levels of expression when *U. maydis* is grown in the presence of L-arabinose [50]. In *S. cerevisiae*, when Ded1 was overexpressed by a factor greater than 10, a slow-growth phenotype and an accumulation of stress granules were observed [26]. A slow-growth phenotype was not observed in the crg1:*uded1* mutants where two copies of *uded1* are present in the genome. Two copies of *uded1* did not increase the transcript level in the crg1:*uded1* mutants (Figure 7) to the degree that it was increased in the Δ*uded1* crg1:*uded1* mutants under permissive conditions. The higher *uded1* transcript levels may be required for inhibiting growth.

When compatible Δ*uded1* crg1:*uded1* mutants are crossed with each other, there is decreased Fuz^+^ formation when compared to wt × wt, wt × Δ*uded1* crg1:*uded1*, or crg1:*uded1* × Δ*uded1* crg1:*uded1* crosses (Figure 6). There was greater Fuz^+^ formation when cultures were grown in a medium containing sorbitol (Figure 6B); however, it was still less dense compared to the wild-type Fuz^+^ formation. Several factors may contribute to the decreased Fuz^+^ formation in the deletion mutant crosses. The mating assay was performed on PDA medium containing charcoal. The primary carbon source in PDA is dextrose, which did not stimulate the crg1 promoter in the deletion mutants and therefore overexpression of *uded1* did not influence growth. The mutants were initially cultured overnight in permissive growth conditions, which stimulated the expression of *uded1* and growth of the haploid cells long enough for fusion of the compatible strains on the PDA plates. The addition of the osmoprotectant sorbitol to the growth medium allowed normal budding growth of the Δ*uded1* crg1:*uded1* mutants. The normal growth contributed to the successful fusion of the compatible strains to form the dikaryon but then the lack of crg1 induction would mean that *uded1* is not expressed and continued growth was inhibited producing a reduced amount of Fuz^+^.

The Δ*uded1* crg1:*uded1* mutants were unable to infect the plant. The inability of the mutants to infect *Z. mays* may be due to insufficient levels of arabinose on the plant surface to ensure ectopic expression of *uded1* from the *ip* locus. The lack of expression would be expected to reduce growth and therefore pathogenesis would not proceed. At the time of infection, the inoculum contained equal concentrations of compatible haploids that are suspended in water. The addition of arabinose to the inoculum was not explored but its addition may have provided sufficient levels of arabinose for *uded1* expression in the deletion mutants which might have allowed dikaryon formation and plant infection.

### 3.3. Uded1’s Role in Translation Regulation

Several studies in the budding yeast show that Ded1 regulates translation [26,28]. Our initial hypothesis was that *uded1* had a role in modulating dsRNA during the exit from teliospore dormancy to germination, possibly aiding the translation of stored mRNAs in the dormant teliospore. Ostrowski and Saville [14] identified *as-ssm1* as a natural antisense transcript to a mitochondrial seryl-tRNA synthetase that is expressed in a teliospore-specific manner. The expression of *as-ssm1* forms an mRNA duplex with *ssm1* that may stabilize the mRNA, repress translation, and prevent degradation during teliospore dormancy [14]. We used the crg1:*uded1* strains to create mutants that expressed the *as-ssm1* transcript from an autonomously replicating vector in haploid cells. We then assessed dsRNA stability in the crg1:*uded1* [pCMas-ssm1] mutants by altering the expression of *uded1* by growing the mutants in repressive and permissive conditions. Under these conditions, *as-ssm1* transcription from the autonomously replicating vector in repressive and permissive conditions was confirmed using RT-PCR (Figure 8). However, the genomic copy of *as-ssm1* was also detected at low levels in the vector-only (crg1:*uded1* [pCM768]) controls in repressive and permissive conditions. Reviewing data from Ostrowski and Saville [14] indicated that the *as-ssm1* transcript is teliospore-specific and is not detected in the SG200 solo-pathogenic *U. maydis* strain. However, transcriptome analysis of the data from Donaldson, Ostrowski, Goulet and Saville [42] indicated low levels of the *as-ssm1* transcript in the wild-type haploid strains 518 and 521. Therefore, detecting *as-ssm1* in the vector-only controls was not unexpected. Under the repressive growth conditions, we found no difference in dsRNA formation when the expression of *uded1* is altered (Figure 10). The inability to detect a difference in dsRNA formation may indicate that the expression of *as-ssm1* from the vector is driving the dsRNA formation to a high level and that the influence of *uded1* cannot be detected above this level. Interestingly, dsRNA formation was detected in the vector-only controls (crg1:*uded1* [pCM768]) when grown in permissive conditions. This detection of an impact of Uded1 on dsRNA formation may be due to the low level of *as-ssm1* in the haploid strains. This suggests that *uded1* may induce conformational changes to the mRNA to stabilize it, and/or bind to *as-ssm1*/*ssm1* to form an mRNP when *as-ssm1* and *ssm1* RNAs are expressed from the genome at lower (wild-type) levels. Therefore, the *uded1* influence on dsRNA formation and the subsequent translation inhibition is linked to RNA levels and/or the site of transcription.

In *S. pombe*, the *uded1* ortholog enhances the antisense RNA gene silencing effect when it is co-expressed with a long antisense. It was suggested that *ded1* functions to stabilize the sense/antisense pair to suppress the expression of the gene [41]. The overexpression of Ded1 in *S. cerevisiae* results in translation repression and the formation of stress granules which contain mRNPs that are stalled in translation [26,27,28]. Ded1 mediates translation when the TOR pathway is inactivated [51] and it was proposed that Ded1 promotes cell survival by repressing general translation which inhibits cell growth and then promoting translation once the stressor has been removed or the cell has adapted [28]. One class of genes that are negative interactors with Ded1 under stress conditions are those involved in mitochondrial translation [52]. Although only nonessential genes were assessed, essential genes, such as the *ssm1* ortholog, may also interact with *DED1*. Given the conserved function of *DED1*/*DDX3*, *uded1* may function to repress the translation of a subset of genes through the formation of mRNPs, especially those formed during teliospore dormancy. Consistent with this, the *uded1* binding to *ssm1*/*as-smm1* to stabilize the dsRNA, observed in our experiments, may result in the creation of an mRNP that stalls translation. The resulting mRNP would likely be sequestered in an RNA granule such as a stress granule where translation could be repressed and the mRNA stabilized [53].

The Δ*uded1* crg1:*uded1* mutants have a mycelial phenotype when grown in permissive conditions. The microscopic analysis (Figure 5) showed a mixture of normal budding cells (cigar-shaped) and elongated cells that appear to be growing in a polarized manner. Forbes et al. [54] reported that cells appear elongated when *ded1* (*sum3*) is overexpressed in *S. pombe*. Overexpression of *ded1* may negatively regulate the cell-cycle response to osmotic stress, possibly interfering with the regulation of proteins in the MAPK pathway [54]. The *uded1* transcript is upregulated in the Δ*uded1* crg1:*uded1* mutants (Figure 7) and may contribute to the mycelial phenotype observed (Figure 5). Normal budding growth is restored in the Δ*uded1* crg1:*uded1* mutants when cells are in the presence of sorbitol, an osmoprotectant [55,56,57] (Figure 5), which suggests that altered *uded1* expression changes the cells so they are sensitive to osmotic stress. The sensitivity in the *uded1* mutants could result from a defect in the cell wall formation or impaired mitotic division.

The *U. maydis uded1* may have further functions similar to its orthologs in other fungi. Aryanpur, Mittelmeier and Bolger [28] reported that overexpression of Ded1 and oxidative stress resulted in growth defects and translational changes. In addition, an accumulation of stress granules in cells overexpressing Ded1 was observed [26]. Our study did not explore the possibility of stress granule accumulation; however, overexpression of *uded1* led to altered phenotypes in response to a stressor and future research could explore any potential link to stress granule formation in *U. maydis*.

Our results suggest a possible function for *uded1* during teliospore dormancy. The RNA-seq analysis from Seto, Donaldson and Saville [23] indicated that the transcript is upregulated in the dormant teliospore. In dormant teliospores, translation is repressed, and we infer that Uded1 levels are increased based on their transcript levels during dormancy. This translation repression may reduce cell growth and promote the entrance into a dormant state. Consistent with this is the slow-growth phenotype observed in our Δ*uded1* crg1:*uded1* mutants where the *uded1* transcript was increased. The proposed function of the *S. pombe* Ded1 is to repress general translation leading to slowed growth with the subsequent reversal of this process in response to oxidative stress changes [28]. We hypothesize that, during teliospore formation, *uded1* is involved in repressing the translation of genes involved in cell proliferation and growth and promoting the translation of genes involved in preparing the teliospore for dormancy. Gene transcripts are stored in the dormant teliospore in the form of mRNPs with Uded1 to stabilize them, possibly in stress granules; then, when the signal for germination is received, Uded1 could unwind the mRNAs and promote translation of the genes involved in cell proliferation and growth resulting in germination and promycelium formation.

In light of the presented research, we view fungal spore dormancy as a stress response. The development of fungal spores involves changes to the cytoplasm, accumulation of protective compounds, and downregulation of cellular processes that involve cell proliferation and growth [58]. These changes aid in protecting the fungus from adverse environmental conditions during liberation and dispersal. Some notable compounds that have been found in dormant fungal spores are trehalose, glycogen, and lipids [1,59,60] which may act as protective compounds and metabolism reserves during dormancy. Fungal spores are typically characterized as having low respiration rates and limited metabolic activity [8,61,62,63] suggesting that many of the genes involved in these biological pathways are translationally repressed. Other general features include thick cell walls, as seen with teliospores, and decreased water content [reviewed in 10]. Enzymes and pre-formed mRNAs essential for growth and metabolism are stored within the dormant fungal spore and are utilized once germination is initiated [8,10,11,61,64,65,66]. Previous work by Donaldson and Saville [13] and Ostrowski and Saville [14] hypothesized that stored mRNAs in the cytoplasm of dormant teliospores were in the form of sense/antisense pairs that are stabilized with an RNA helicase such as *uded1* to form an mRNP. Teliospore germination is induced under favourable growth conditions, which we interpret as equivalent to removing the stress that the spore senses [58]. The RNA helicases we examined, *udbp3* and *uded1*, are upregulated in the dormant teliospore. Our research has uncovered that these RNA helicases are involved in responding to stresses through interacting with RNA metabolism. Specifically, *udbp3* is involved in regulating stress-responsive transcription factors or genes and *uded1* is involved in slowing growth and interacting with dsRNA also in response to stress. We propose that the *uded1* effect on growth is the result of translation repression and promotion during teliospore dormancy, and that Uded1 may function to bind and stabilize sense/antisense transcripts for storage in the dormant teliospore and then reactivate the mRNAs upon germination. Viewing the formation and germination of teliospores as a stress response involving the activity of these helicases increases our understanding of the molecular events involved during teliospore formation and germination and may aid in developing novel methods for mitigating fungal disease spread and progression.

## 4. Materials and Methods

### 4.1. Ustilago maydis Strains and Growth Conditions

The *U. maydis* strains used in this study are listed in Table 1. Unless otherwise noted, all *U. maydis* strains are grown in YEPS medium (1% *w*/*v* yeast extract, 2% *w*/*v* peptone, 2% *w*/*v* sucrose) at 28 °C, shaking at 250 rpm. Mutant strains that contain the hygromycin resistance cassette are grown on medium supplemented with 250 µg/mL hygromycin B (Bioshop Canada, Burlington, ON, Canada) and strains that contain the carboxin resistance cassette are grown on medium supplemented with 4 µg/mL carboxin (MililporeSigma, Oakville, ON, Canada).

### 4.2. Bioinformatic Analysis

*Ustilago maydis* RNA helicases were previously identified by Seto and Saville [29]. The transcriptome data from Seto, Donaldson and Saville [23] was utilized to identify RNA helicases with transcript levels that were upregulated in the dormant teliospore and decreased during germination.

Identification of the RNA helicase core for Udbp3 and Uded1 was carried out by aligning protein sequences against orthologs identified by Seto and Saville [29]. Protein sequences were aligned in Jalview v2.11.2.0 [31] using the JABAWS service to perform the MUSCLE alignment under default settings [30]. The RNA helicase core sequence motifs for all RNA helicases in *S. cerevisiae* were previously compared to other orthologs in Fairman-Williams, Guenther and Jankowsky [20] and was used to identify the RNA helicase core in Udbp3 and Uded1. The aligned protein sequences were used to construct maximum likelihood phylogenetic trees using W-IQ-TREE multicore version 1.6.12 [32]. Default settings were selected, 1000 ultrafast bootstrap alignments, and approximate Bayes test were conducted. Phylogenetic trees were visualized and analyzed using FigTree v1.4.4 (available online: http://tree.bio.ed.ac.uk/software/figtree/, accessed 10 February 2023).

### 4.3. Creation of udbp3 Deletion Strains

Deletion constructs were created for *udbp3* utilizing the PCR-based method outlined in Kämper [35]. The plasmids pMF1-hs [35] and pMF1-c [68] (Appendix A) were used to obtain the hygromycin (*hph*^R^) and the carboxin resistance cassettes (*cbx*^R^). Both resistance cassettes were PCR amplified from linearized plasmid DNA using the primers HygCarb_Out_pMF1-F and HygCarb_Out_pMF1-R (Appendix A). The upstream and downstream flanking regions of the *udbp3* ORF were PCR amplified using primers Dbp3_LF-F and Dbp3_LF_SfiI-R(2) for the 5′ flank and Dbp3_RF_SfiI(2) and Dbp3_RF-R(2) for the 3′ flank (Appendix A) to introduce the *Sfi*I restriction endonuclease cleavage site. All PCR products were purified using the PureLink PCR Purification Kit (Thermo Fisher Scientific, Mississauga, ON, Canada), digested with *Sfi*I (New England Biolabs, Whitby, ON, Canada), and gel purified with the PureLink Quick Gel Extraction Kit (Thermo Fisher Scientific, Mississauga, ON, Canada). Equal concentrations of the purified flanking regions and resistance cassette were ligated using T4 DNA ligase (New England Biolabs, Whitby ON, Canada) and were purified using the PureLink Quick Gel Extraction Kit (Thermo Fisher Scientific, Mississauga, ON, Canada). The Hyg^R^ and Carb^R^ constructs were cloned into the pCR2.1 TOPO vector and transformed into One Shot TOP10 competent *E. coli* cells (Thermo Fisher Scientific, Mississauga, ON, Canada). Blue-white screening and kanamycin selection were utilized to select putative transformants containing the *udbp3* deletion construct. Putative bacterial transformants were cultured in LB Broth (Miller) (MilliporeSigma, Oakville, ON, Canada) containing 100 µg/mL ampicillin (BioShop Canada, Burlington, ON, Canada) overnight and plasmid DNA was isolated using the PureLink Quick Plasmid Miniprep Kit (Thermo Fisher Scientific, Mississauga, ON, Canada). Successful transformants were confirmed through sequencing using primers in Appendix A using BigDye Terminator chemistry V.3.1 (Thermo Fisher Scientific, Mississauga, ON, Canada) and an automated sequencer (ABI 3730 DNA analyzer, Thermo Fisher Scientific, Mississauga, ON, Canada). Raw sequences were trimmed and assembled in SeqMan Pro V.11.2.1 using default settings and aligned using MEGA 7 [69].

Confirmed deletion constructs were PCR amplified from pCR2.1TOPOΔUMAG_01732-HygR and pCR2.1TOPOΔUMAG_01732-CarbR using primers Dbp3_LF_Nested-F and Dbp3_RF-R (Appendix A) and Phusion High-Fidelity DNA polymerase (Thermo Fisher Scientific, Mississauga, ON, Canada). PCR products were gel purified with the PureLink Quick Gel Extraction Kit (Thermo Fisher Scientific, Mississauga, ON, Canada). The Hyg^R^ deletion construct was transformed into 518 competent protoplasts and the Carb^R^ deletion construct was transformed into 521 competent protoplasts to replace *udbp3* with the resistance cassette through homologous recombination. Competent protoplasts and *U. maydis* transformation was performed using the method of Wang et al. [70] with modifications from Morrison et al. [71]. Transformants were plated on DCM containing 2.0% *w*/*v* D-glucose (BioShop Canada, Burlington, ON, Canada), 1 M sorbitol (BioShop Canada, Burlington, ON, Canada), and supplemented with either 250 µg/mL hygromycin B or 4 µg/mL carboxin. Genomic DNA was isolated from successful transformants using the method described by Hoffman and Winston [72]. Putative transformants were screened with PCR and confirmed through Southern blot analysis. Southern blot analysis was conducted using the DIG High Prime DNA labelling and Detection Kit 1 (MilliporeSigma, Oakville, ON, Canada). Probes were created for the hygromycin and carboxin resistance cassettes following the manufacturer’s protocol.

### 4.4. Creation of an uded1 Ustilago maydis Ectopic Expression Strain

The creation of the ectopic expression construct followed the methods outlined in Ostrowski and Saville [14] which used the p123 shuttle vector (Appendix A). The p123 vector contains ampicillin resistance for bacterial transformation and carboxin resistance targeted to *U. maydis* genome integration at the *ip* locus. The p123 vector contains the constitutive otef promoter (P_otef_) upstream of the GFP reporter (*egfp*) [73].

The *uded1* expression construct was created to express *uded1* under an L-arabinose inducible promoter. The p123 vector was first modified by replacing P_otef_ with the crg1 promoter (P_crg1_). P_crg1_ was PCR amplified from pMF2-1h [68] (Appendix A) with primers crg1-KpnI-F and crg1-XmaI-R (Appendix A) using Phusion High-Fidelity DNA Polymerase (Thermo Fisher Scientific, Mississauga, ON, Canada). These primers introduced the restriction endonuclease cleavage sites *Kpn*I and *Xma*I at the 5′ and 3′ ends, respectively, for insertion into p123 in place of P_otef_. The PCR product and p123 were digested with *Kpn*I and *Xma*I (New England Biolabs, Whitby, ON, Canada) and purified with the PureLink Quick Gel Extraction Kit (Thermo Fisher Scientific, Mississauga, ON, Canada) following the manufacturer’s protocols. The digested and purified PCR product and p123 without P_otef_ were ligated with T4 DNA ligase, transformed into Subcloning Efficiency DH5α Competent *Escherichia coli* cells (Thermo Fisher Scientific, Mississauga, ON, Canada), and plated on LB Broth (Miller) (MilliporeSigma, Oakville, ON, Canada) supplemented with 100 µg/mL ampicillin following the manufacturer’s protocols. Bacterial colonies were cultured overnight in 3.0 mL of LB Broth (Miller) (MilliporeSigma, Oakville, ON, Canada) supplemented with 100 µg/mL ampicillin and plasmid DNA was isolated using the PureLink Quick Plasmid Miniprep Kit (Thermo Fisher Scientific, Mississauga, ON, Canada) following the manufacturer’s protocols. Putative transformants were verified by sequencing following the protocols outlined above and using the plasmid DNA.

The *uded1* expression construct was created using the modified p123 vector (p123 + crg1). *uded1* was PCR amplified with primers UMAG_04080_NcoI-F and UMAG_04080_NotI_R from *U. maydis* gDNA using Phusion High Fidelity DNA Polymerase. Both p123 + crg1 and PCR products were digested with *Nco*I-HF and *Not*I-HF (New England Biolabs, Whitby, ON, Canada). Digested products were gel purified with the PureLink Quick Gel Extraction Kit (Thermo Fisher Scientific, Mississauga, ON, Canada), followed by ligation with T4 DNA ligase (New England Biolabs, Whitby, ON, Canada). The ligated product produced the p123 + crg1 + *uded1* construct which was transformed into Subcloning Efficiency DH5α Competent *Escherichia coli* cells (Thermo Fisher Scientific, Mississauga, ON, Canada) and grown on LB Broth (Miller) (MilliporeSigma, Oakville, ON, Canada) plates supplemented with 100 µg/mL ampicillin with modifications to the manufacturer’s protocols. Following transformation, the plated transformants were incubated at 28 °C for two days to promote slow growth of *E. coli*. Previous transformations at the recommended 37 °C incubation temperature yielded little to no transformants which suggested leaky expression of *uded1* from the plasmid resulting in the gene being toxic to *E. coli*. Bacterial colonies were cultured overnight at 28 °C in 3.0 mL of LB Broth (Miller) (Thermo Fisher Scientific, Mississauga, ON, Canada) supplemented with 100 µg/mL ampicillin and plasmid DNA was isolated using the PureLink Quick Plasmid Miniprep Kit (Thermo Fisher Scientific, Mississauga, ON, Canada) following the manufacturer’s protocols. The construct was verified through sequencing using the procedure above using primers listed in Appendix A.

The confirmed construct was linearized with *Ssp*I (New England Biolabs, Whitby, ON, Canada) and purified with the PureLink PCR Purification Kit (Thermo Fisher Scientific, Mississauga, ON, Canada). The linearized construct was transformed into competent *U. maydis* 518 and 521 protoplasts for integration into the *ip* locus using the transformation protocols described above. The transformed protoplasts were plated on DCM plates containing 2.0% *w*/*v* D-glucose and 1 M sorbitol and supplemented with 4 µg/mL carboxin. Genomic DNA was isolated from successful transformants using the method described above. The transformants were screened for multiple p123 + crg1 + *uded1* insertions with PCR using the primers p123multi-F and p123multi-R (Appendix A). Putative transformants that did not PCR amplify with these primers passed the PCR screen. Transformants that passed this screen were PCR amplified with primers umgapd-F and umgapd-R (Appendix A) to ensure genomic DNA was amplifiable. The putative transformants were confirmed with Southern blot analysis using the DIG High Prime DNA labelling and Detection Kit 1 (MilliporeSigma, Oakville, ON, Canada) using the carboxin resistance-specific probe.

### 4.5. Creation of an uded1 Ustilago maydis Deletion Strain

Viable mutants could not be obtained using the traditional PCR-based method [35] to delete *uded1* suggesting that deletion of *uded1* is lethal. Instead, the two-step gene disruption method employed by Ostrowski and Saville [14] was utilized to delete the native *uded1* from its native locus in the ectopic expression strains (518 crg1:*uded1* and 521 crg1:*uded1*).

The *uded1* deletion construct was created following the methods in Kämper [35]. The Hyg^R^ cassette was PCR amplified with the primers HygCarb_Out_pMF1-F and HygCarb_Out_pMF1-R (Appendix A) from linearized pMF1-hs DNA. The 5′ and 3′ flanking regions of *uded1* were PCR amplified from *U. maydis* genomic DNA with the primers Ded1_LF-F and Ded1_LF_SfiI-R(2), and Ded1_RF_SfiI-F(2) and Ded1_RF-R (Appendix A) respectively to introduce an *Sfi*I restriction endonuclease cleavage site. All PCR products were PCR purified, digested with *Sfi*I, and purified by gel extraction. Equal concentrations of the purified Hyg^R^ cassette, *uded1* 5′ flanking region, and *uded1* 3′ flanking region were ligated using T4 DNA ligase (New England Biolabs, Whitby, ON, Canada). The ligated product (~3.9 kb) was gel extracted and purified following the same method for creating the *udbp3* deletion mutants. The purified ligation product was cloned into the pCR 2.1 TOPO vector and transformed into One Shot TOP10 competent *E. coli* cells (Thermo Fisher Scientific, Mississauga, ON, Canada). Putative transformants were selected using blue-white screening and kanamycin selection. Putative transformants were confirmed by sequencing as outlined above using primers listed in Appendix A.

The confirmed deletion construct was PCR amplified from the pCR2.1TOPOΔ*uded1*-HygR plasmid using nested primers Ded1_LF_Nest-F and Ded1_RF_Nest-R (Appendix A) and Phusion High-Fidelity DNA polymerase (Thermo Fisher Scientific, Mississauga, ON, Canada). The PCR product was gel extracted and purified as described above. The purified deletion construct was transformed into 518 crg1:*uded1* and 521 crg1:*uded1* competent protoplasts to replace *uded1* at its native locus by homologous recombination. Transformants were plated on DCM plates containing 1.0% *w*/*v* L-arabinose (MilliporeSigma, Oakville, ON, Canada), 1 M sorbitol, and supplemented with 250 µg/mL hygromycin B. Medium containing L-arabinose ensured ectopic expression of *uded1* at the *ip* locus to allow for growth of transformants. Genomic DNA was isolated from putative transformants following the methods above and screened by PCR using the primers 04080_LF-Seq-F1 and 04080-Seq-R1 (Appendix A). Putative transformants that passed the PCR screen were confirmed through Southern blot analysis using the methods outlined above. The hygromycin resistance cassette was probed following the manufacturer’s protocol.

### 4.6. Creation of as-ssm1-Expressing Strains

The *ssm1* antisense (*as-ssm1*) was previously characterized by Ostrowski and Saville [14]. Donaldson and Saville [13] created an expression vector to express *as-ssm1* from an autonomously replicating sequence in the pCM768 vector. The pCM[as-ssm1] vector was transformed into competent 518 crg1:*uded1* and 521 crg1:*uded1* protoplasts using the García-Pedrajas et al. [74] *U. maydis* transformation method. All transformed cells were plated and grown on YEPS plates supplemented with 250 µg/mL hygromycin B. DNA was isolated from putative transformants and PCR screened with primers pGAP(-79)Forward and Um12232 PCR-F to amplify the antisense transcript and HYG-Seq-F2 and HYG-Seq-R3 which amplify the Hyg^R^ cassette (Appendix A) to verify successful *U. maydis* transformation. Furthermore, antisense transcript expression was confirmed via RT-PCR for all transformants.

### 4.7. Total RNA Isolation, RT-PCR, and RT-qPCR

Total RNA was isolated from all samples, DNaseI-treated, and screened for gDNA contamination following the methods outlined in Doyle et al. [75] and Morrison, Donaldson and Saville [71]. RNA was quantified with a NanoDrop 8000 Spectrophotometer (Thermo Fisher Scientific, Mississauga, ON, Canada) and quality was assessed by electrophoretic separation of glyoxalated RNA on an agarose gel using methods outlined in Sambrook and Russell [76]. Reverse transcription was carried out on 200 ng of DNase I-treated RNA in a 10 µL reaction using TaqMan Reverse Transcription Reagents (Thermo Fisher Scientific, Mississauga, ON, Canada). For RT-PCR, RNA was primed with oligo-d(T)_16_ and for strand-specific first-strand synthesis reactions, RNA was primed with primers listed in Appendix A. The oligo-d(T)_16_ reactions were carried out under the following conditions: 25 °C for 10 min, 50 °C for 30 min, 95 °C for 10 min, followed by a 4 °C hold. Strand-specific first-strand synthesis reactions were carried out under the conditions outlined by Ho, Donaldson and Saville [43]. Following first-strand synthesis, the cDNA was diluted fourfold (1:3) with dH_2_O.

Primers for all RT-PCR reactions were designed in Primer3 v4.1.0 [77,78,79] and are listed in Appendix A. All RT-PCR reactions were performed with 2.0 µL of diluted cDNA and DreamTaq DNA Polymerase (Thermo Fisher Scientific, Mississauga, ON, Canada). PCR cycling conditions were: 95 °C for 3 min, then 35 cycles of 95 °C for 30 s, 60 °C for 30 s, and 72 °C for 1 min, then 72 °C for 10 min, followed by a 4 °C hold. One-third of the RT-PCR product was separated by agarose gel electrophoresis. Agarose gels contained 0.3 µg/mL of ethidium bromide (BioShop Canada, Burlington, ON, Canada), and PCR products were visualized under UV light.

Primers and TaqMan MGB probes for RT-qPCR were designed with Primer Express Software version 2.0 (Thermo Fisher Scientific, Mississauga, ON, Canada) using default criteria for genes *UMAG_00175* and *uded1* (Appendix A). RT-qPCR reactions were carried out with 2.0 µL of diluted cDNA in a 20 µL reaction with TaqMan Universal PCR Master Mix (Thermo Fisher Scientific, Mississauga, ON, Canada) on the QuantStudio 3 Real-Time PCR System (Thermo Fisher Scientific, Mississauga, ON, Canada). The RT-qPCR cycling conditions were: 50 °C for 2 min, 95 °C for 10 min, followed by 40 cycles of 95 °C for 15 s and 60 °C for 1 min. Data were collected and analyzed using the QuantStudio Design & Analysis Software version 2.6.0 (Thermo Fisher Scientific, Mississauga, ON, Canada). The comparative C_T_ (2^−ΔΔCT^) method was used to analyze the results where *UMAG_00175* was set as the endogenous control. The calibrator control is specified for each experiment.

### 4.8. S1 Nuclease Protection Assay

RNA was isolated from *as-ssm1* expressing strains that were grown and harvested from YEPS and YEPA (1% *w*/*v* yeast extract, 2% *w*/*v* peptone, 1% *w*/*v* L-arabinose) plates supplemented with 250 µg/mL hygromycin B. Cells were frozen in liquid nitrogen and stored at −80 °C prior to RNA extraction. The frozen cells were resuspended in TRIzol reagent (Thermo Fisher Scientific, Mississauga, ON, Canada) and transferred to 2.0 mL screw-cap tubes containing Lysing Matrix C (MP Biomedicals, Santa Ana, CA, USA). Cells were disrupted and RNA was isolated following the methods described above. All RNA samples were precipitated, DNase I-treated, and screened for gDNA contamination as described above.

S1 nuclease digestion reactions were carried out on 2.5 µg of DNase I-treated RNA and incubated at 37 °C for 30 min. Each reaction contained the following final concentration of S1 nuclease (Thermo Fisher Scientific, Mississauga, ON, Canada): 0, 0.01, 0.1, or 1 U/µL. The dsRNA was extracted by phenol/chloroform, precipitated with NH_4_Ac/Ethanol/GlycoBlue Coprecipitant (Thermo Fisher Scientific, Mississauga, ON, Canada) at −20 °C for at least 60 min, and resuspended in 15 µL DEPC-treated water. Strand-specific first-strand synthesis was performed on 2.0 µL of S1 trimmed RNA using tagged *ssm1* sense-specific first-strand primers that targeted either the overlapping region of the sense/antisense region (um12232_FS_Sense_Tag) or the non-overlapping region (um12232_FS_S_NO) (Appendix A). The RT-PCRs were carried out targeting the overlapping and non-overlapping regions to determine dsRNA protection and PCR cycling was performed using 40 cycles instead of 35.

### 4.9. Plate Mating Assays

Plate mating assays were carried out by a modified method of Donaldson et al. [80]. The *udbp3* mutant strains were grown overnight in YEPS and the *uded1* mutant strains were grown overnight in liquid DCM containing 1.0% *w*/*v* L-arabinose and DCM containing 1.0% *w*/*v* L-arabinose and 1 M sorbitol. All cells were washed and diluted to an OD_600_ of 1.0 with sterile water. Equal volumes of compatible haploids were pre-mixed and 5.0 µL were spotted on PDA (BioShop Canada, Burlington, ON, Canada) plates containing 1.0% *w*/*v* activated charcoal (MilliporeSigma, Oakville, ON, Canada). All plates were incubated at room temperature and filamentous growth was assessed 2–3 days post spotting.

### 4.10. Seedling Pathogenesis Assays

Pathogenesis assays on Golden Bantam *Z. mays* seedlings were performed following the protocols outlined in Morrison, Donaldson and Saville [71]. Disease symptoms were scored at 7, 10, and 14 days post-infection using the scoring system outlined in Cheung et al. [81]. All pathogenesis assays were performed in triplicate with approximately 45 plants per biological replicate. Statistical analysis was performed as outlined in Cheung, Donaldson, Storfie, Spence, Fetsch, Harrison and Saville [81].

### 4.11. Stress Response Assays

The osmotic stress response caused by NaCl was assessed in the *udbp3* deletion mutants. All strains were cultured overnight in YEPS and normalized to an OD_600_ of 1.0. Cells were pelleted, washed with sterile dH_2_O, and resuspended in sterile dH_2_O. A ten-fold dilution series was created for each strain and 5.0 µL of the dilution series was plated on YEPS plates containing 1 M NaCl and Minimal Medium (MM) plates containing 1.0% *w*/*v* D-glucose and 1 M NaCl. Plates were incubated at 28 °C for three days.

### 4.12. Plate Growth Assays

Mycelial growth of the *uded1* mutant strains was assessed by streaking all strains on DCM plates containing 1.0% *w*/*v* L-arabinose, 1 M sorbitol, and antibiotic selection. All plates were incubated at 28 °C for four days. For each *U. maydis* strain, single colonies were picked and streaked on YEPS and YEPA plates. Plates were incubated at 28 °C for three days. The growth on the plate was observed using a Leica S8 APO stereo microscope (Leica Microsystems, Concord, ON, Canada). Photographs were obtained using the Leica EC3 camera and images were analyzed using the Leica Application Suite EZ software v2.1.0 (Leica Microsystems, Concord, ON, Canada). A small portion from the middle of the growth on the plate was picked to include both budding and filamentous cells and was resuspended in 100 µL sterile dH_2_O. Cells were observed using the AxioScope.A1 compound microscope (Carl Zeiss MicroImaging, Toronto, ON, Canada) and differential interference contrast (DIC) images were created at 400× magnification using an Axiocam 208 colour camera (Carl Zeiss MicroImaging, Toronto, ON, Canada).

## Figures and Tables

**Figure 1 ijms-26-02432-f001:**
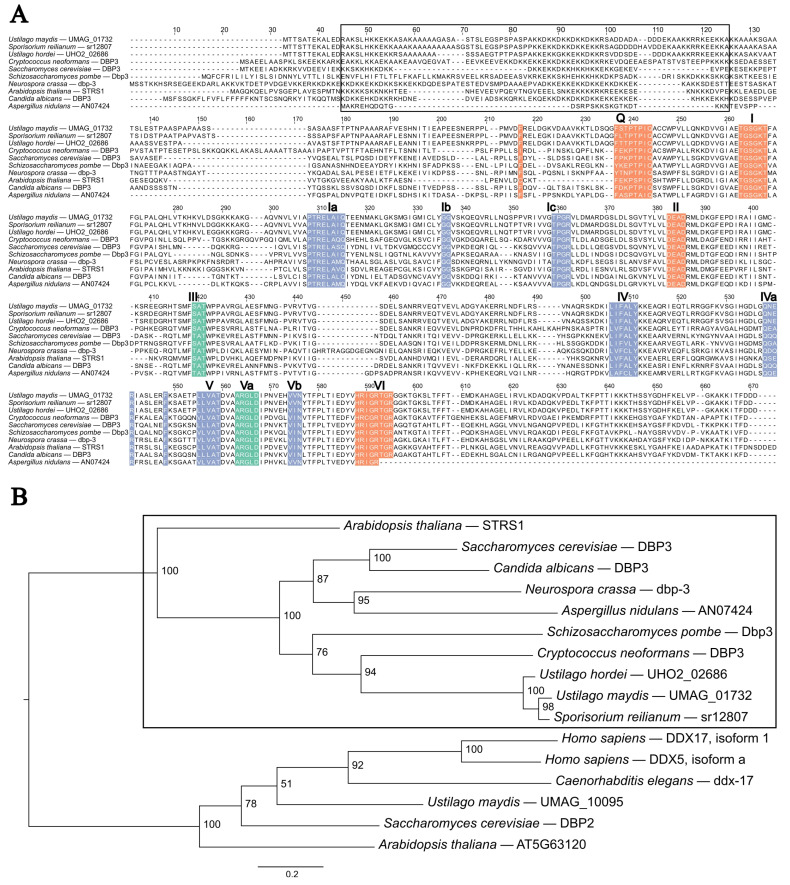
Protein sequence alignment and maximum likelihood phylogenetic tree for Udbp3 and its orthologs. (**A**) Protein sequence alignment of Udbp3 and putative orthologs. MUSCLE alignment [30] of protein sequences was performed and visualized in Jalview v2.11.2.0 [31]. The repetitive KKX sequence motif is identified by the black box. The RNA helicase core sequence motifs, indicated by bold letters and numbers, were identified based on motif identification from Fairman-Williams, Guenther and Jankowsky [20]. The coloured boxes indicate the sequence motifs based on their predominant biochemical function: Orange, ATP binding and hydrolysis; Blue, nucleic acid binding; Green, coordination of NTP and nucleic acid binding site. (**B**) Maximum likelihood phylogenetic tree of Udbp3 orthologs was created using W-IQ-Tree multicore version 1.6.12 with default settings [32], 1000 ultrafast bootstrap alignments, and approximate Bayes test. The tree was visualized using FigTree v1.4.4 (available online: http://tree.bio.ed.ac.uk/software/figtree/, accessed on 10 February 2023). The tree was rooted at the midpoint, the bootstrap value is indicated for each node, and the scale bar indicates the expected number of substitutions per amino acid. The protein for each organism is indicated and the black box indicates the clade of Udbp3 orthologs.

**Figure 2 ijms-26-02432-f002:**
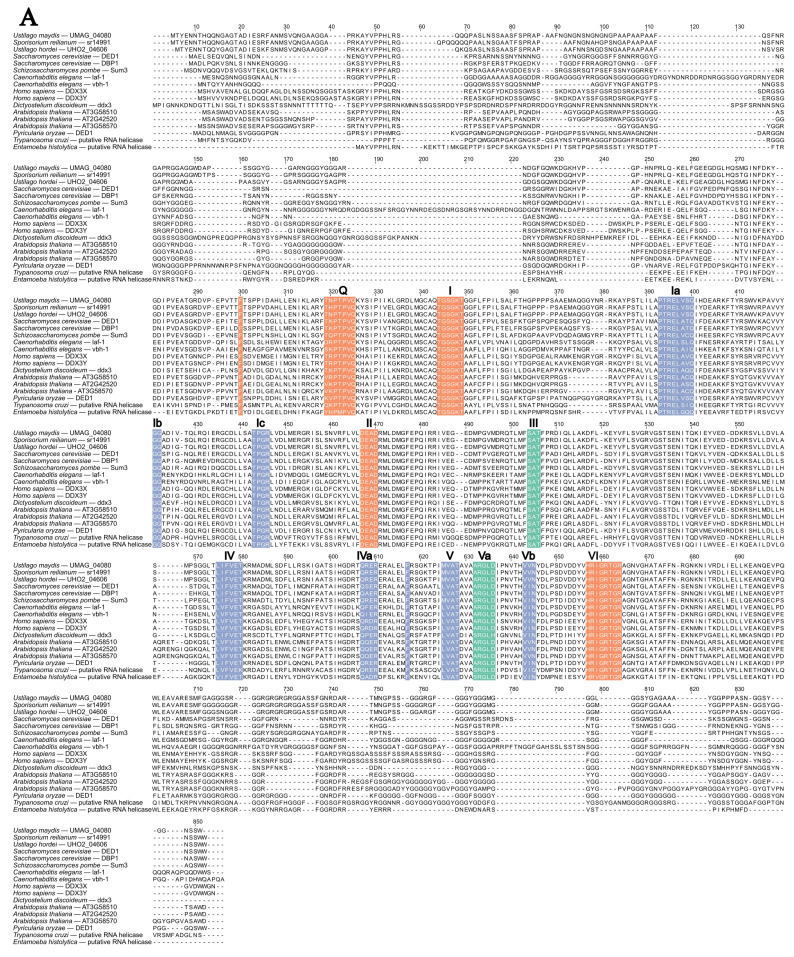
Uded1 protein sequence alignment and maximum likelihood phylogenetic tree. (**A**) MUSCLE alignment [30] of Uded1 and putative orthologs. The RNA helicase core sequence motifs, indicated in bold letters and numbers, were identified based on analysis from Fairman-Williams, Guenther and Jankowsky [20]. The coloured boxes indicate sequence motif based on their predominant biochemical function: Orange, ATP binding and hydrolysis; Blue, nucleic acid binding; Green, coordination of NTP and nucleic acid binding site. (**B**) Maximum likelihood phylogenetic tree of Uded1 orthologs was constructed using W-IQ-Tree with default settings [32], 1000 ultrafast bootstrap alignments, and approximate Bayes test. The tree was visualized using FigTree v1.4.4 (available online: http://tree.bio.ed.ac.uk/software/figtree/, accessed on 10 February 2023). The tree was rooted at the midpoint, the bootstrap value is indicated for each node, and the scale bar indicates the expected number of substitutions per amino acid. The name of the protein for each organism is indicated and the box indicates the clade of Uded1 orthologs.

**Figure 3 ijms-26-02432-f003:**
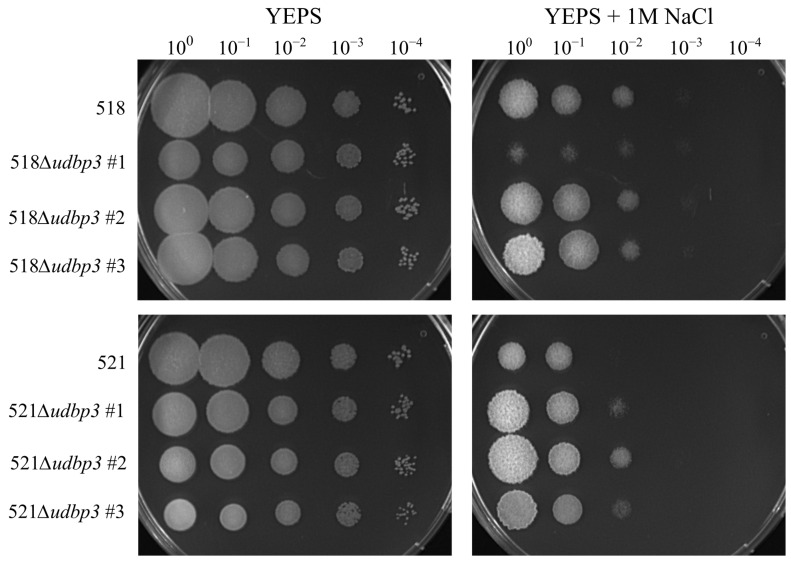
The effects of osmotic stress on Δ*udbp3* mutants. Overnight cultures were normalized to an OD_600_ = 1.0 and a 10-fold serial dilution series was created for all strains. All *U. maydis* cultures were spotted on YEPS (control plate) and YEPS containing 1 M NaCl. Plates were incubated at 28 °C and photographed after 3 days. The data shown are representative of three technical replicates of the spotting assay.

**Figure 4 ijms-26-02432-f004:**
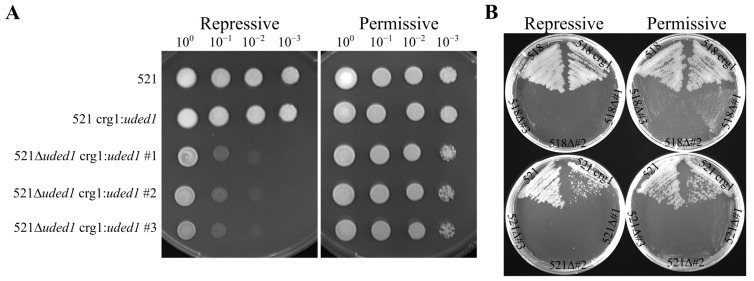
Growth of *uded1* mutants in permissive and repressive growth conditions. (**A**) A 10-fold serial dilution series was plated on MM containing 1.0% D-glucose (repressive) and MM containing 1.0% L-arabinose (permissive). Plates were incubated at 28 °C and photographs of each plate were taken after 3 days. The data shown is representative of three technical replicates of the spotting assay. (**B**) *uded1* mutants streaked onto YEPS (repressive) and YEPA (permissive) plates and incubated at 28 °C. Photographs were taken at 7 days post-incubation. Label abbreviations: crg1 delineates the crg1:*uded1* strain and Δ#1–3 delineates the Δ*uded1* crg1:*uded1* biological replicates.

**Figure 5 ijms-26-02432-f005:**
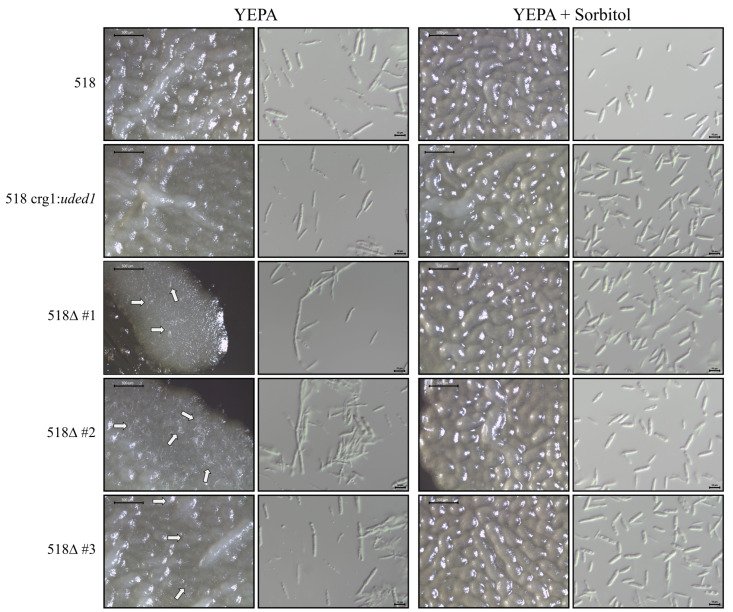
The effects of sorbitol addition to solid medium on the growth of *uded1* mutants. Single colonies of each *U. maydis* strain were streaked onto YEPA and YEPA containing 1 M sorbitol plates and incubated at 28 °C for 3 days. Microscopic images were taken of the growth on the plate (40×) and of the cells resuspended in sterile dH_2_O (400× magnification). Scale bar indicates 500 µm (plate micrographs) and 10 µm (cell micrographs). Data shown is representative of three technical replicates of the growth assay. The results of the growth assay in the 521 *uded1* mutants is shown in Appendix A. Arrows indicate mycelial growth. Label abbreviations: Δ#1–3 delineates the Δ*uded1* crg1:*uded1* biological replicates.

**Figure 6 ijms-26-02432-f006:**
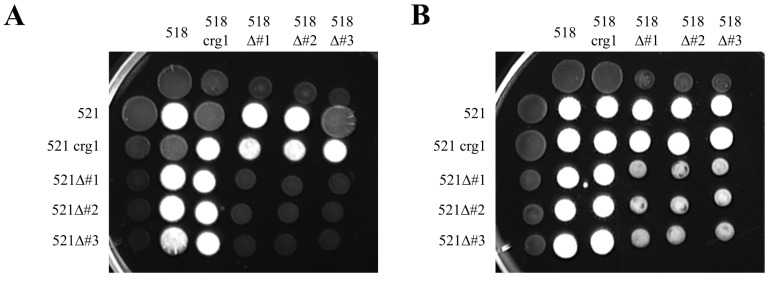
Mating assay of *uded1* mutants. (**A**) *Ustilago maydis* strains were cultured overnight in DCM containing 1.0% L-arabinose. (**B**) *Ustilago maydis* strains were cultured overnight in DCM containing 1.0% L-arabinose and 1 M sorbitol. All cultures were normalized, and compatible strains were premixed. Premixed cultures were spotted on PDA containing 1.0% activated charcoal. Plates were incubated at room temperature. Photos were taken after 3 days and the representative data of three technical replicates of the mating assay is shown. The label abbreviations are as follows: crg1 indicates the crg1:*uded1* mutants and Δ#1–3 indicates the Δ*uded1* crg1:*uded1* mutants.

**Figure 7 ijms-26-02432-f007:**
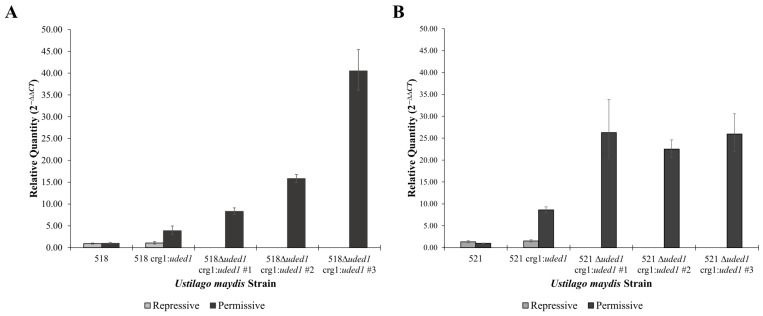
The *uded1* transcript is upregulated in the deletion strains when grown in permissive conditions. The relative quantities were determined by RT-qPCR and calculated using the comparative C_T_ (2^−ΔΔCT^) method, *UMAG_00175* was the endogenous control, and the parent strain (518 or 521) grown in repressive conditions was set as the calibrator. (**A**) 518 wild-type and mutant strains. (**B**) 521 wild-type and mutant strains. Bars indicate the RQ minimum and RQ maximum values (95% confidence interval, n = 3).

**Figure 8 ijms-26-02432-f008:**
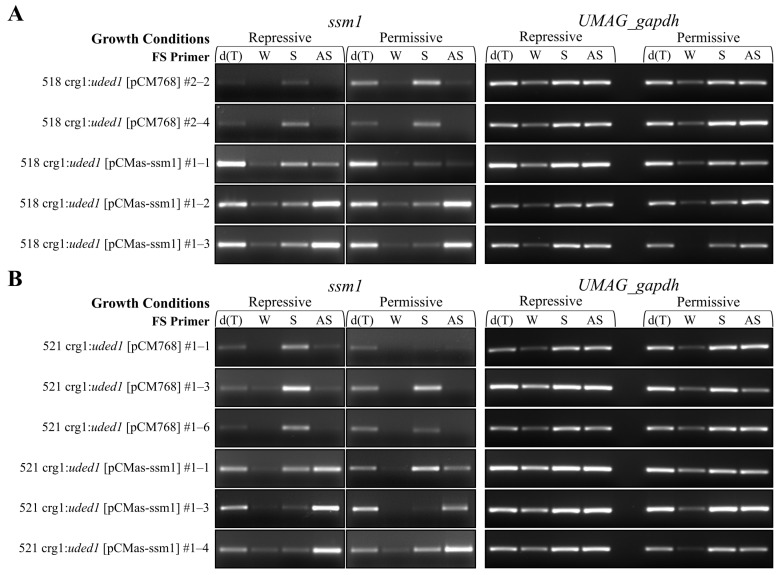
RT-PCR detection of the *ssm1* and *as-ssm1* transcripts in crg1:*uded1* mutants where *as-ssm1* is expressed from an autonomously replicating vector. (**A**) *as-ssm1* expressed in the 518 crg1: *uded1* mutant strains. (**B**) *as-ssm1* expressed in the 521 crg1:*uded1* mutant strain. RNA sources include crg1:*uded1* [pCM768] controls and crg1:*uded1* [pCMas-ssm1] strains grown in YEPS (repressive) and YEPA (permissive) medium. First-strand cDNA synthesis primers include Oligo(dT)_16_ (dT), DEPC-treated H_2_O (W), *ssm1* sense-specific primer (S), and *as-ssm1* antisense-specific primer (AS). *UMAG_gapdh* was used as the housekeeping gene, a control.

**Figure 9 ijms-26-02432-f009:**
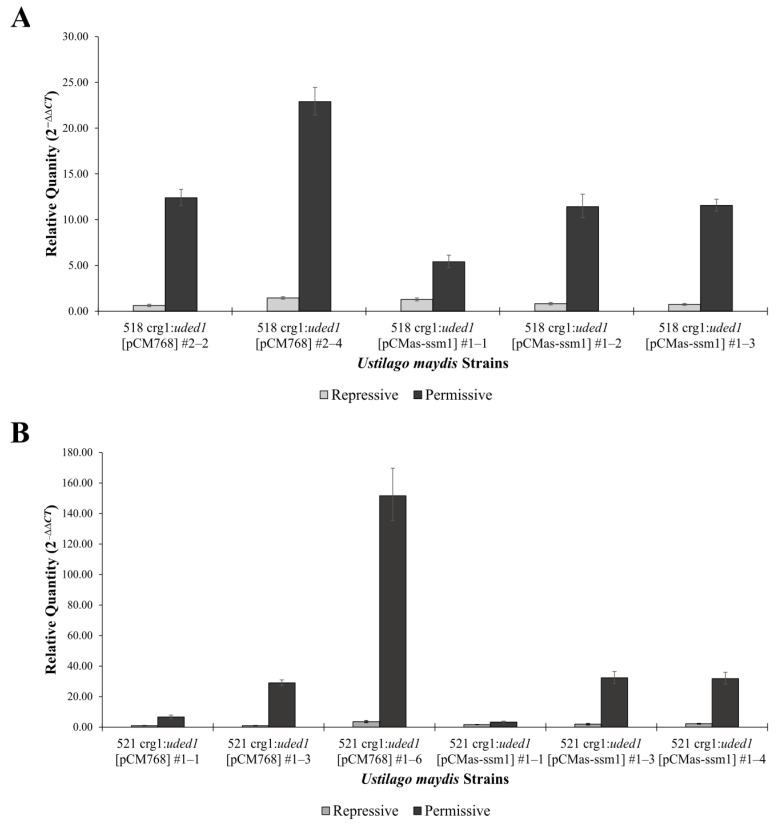
The *uded1* transcript is upregulated in the crg1:*uded1* mutants grown in permissive conditions. All *U. maydis* strains were grown in YEPS (repressive) and YEPA (permissive) medium. Relative quantities were determined by RT-qPCR and calculated using the comparative C_T_ (2^−ΔΔCT^) method, *UMAG_00175* was the endogenous control, and the parent strain (crg1:*uded1*) grown in repressive conditions was set as the calibrator. Assessment of the *uded1* transcript level in the (**A**) 518 crg1:*uded1* [pCMas-ssm1] mutants and (**B**) 521 crg1:*uded1* [pCMas-ssm1]. Bars indicate the RQ minimum and RQ maximum values (95% confidence interval, n = 3).

**Figure 10 ijms-26-02432-f010:**
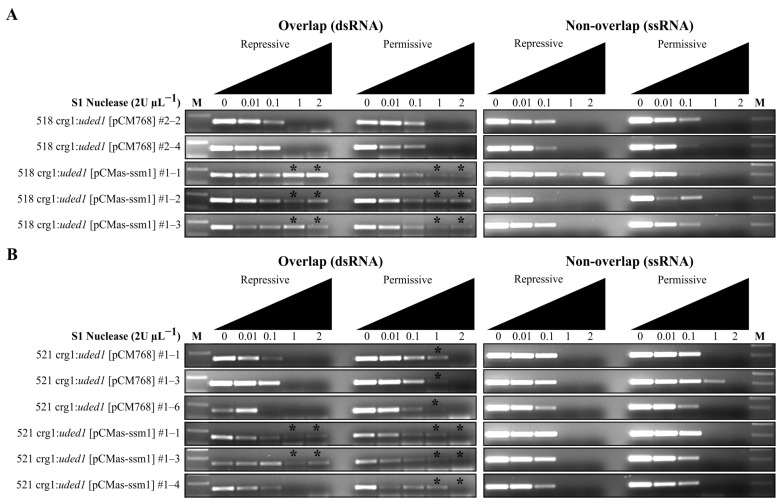
S1 nuclease protection assay of *ssm1* and *as-ssm1* interactions. The (**A**) 518 crg1:*uded1* [pCMas-ssm1] and (**B**) 521 crg1:*uded1* [pCMas-ssm1] mutants were grown in repressive (YEPS) and permissive (YEPA) medium. Increasing amounts of S1 nuclease were used to digest equal quantities of RNA. RNA sources include crg1:*uded1* [pCM768] controls and crg1:*uded1* [pCMas-ssm1] strains. A tagged *ssm1* sense-specific primer was used to generate cDNA. A *gapdh*-specific first-strand synthesis primer was included and used as an internal control to assess *UMAG_gapdh* transcript levels. A DNA molecular weight marker (M) was included. The asterisk (*) indicates increased S1 nuclease digestion resistance. The lighting of the gel images was adjusted using Inkscape v.1.3.2.

**Table 1 ijms-26-02432-t001:** The *Ustilago maydis* strains used in this study.

Strain	Relevant Genotype *	Source
Wild-type		
518	*a2 b2*	Holliday [67]
521	*a1 b1*	Holliday [67]
*uded1* Mutants		
518 crg1:*uded1*	*a2 b2* crg1:*uded1*::*cbx^R^*	This study
521 crg1:*uded1*	*a1 b1* crg1:*uded1*::*cbx^R^*	This study
518Δ*uded1* crg1:*uded1*	*a2 b2* Δ*uded1*::*hph^R^* crg1:*uded1*::*cbx^R^*	This study
521Δ*uded1* crg1:*uded1*	*a1 b1* Δ*uded1*::*hph^R^* crg1:*uded1*::*cbx^R^*	This study
518 crg1:*uded1* [pCM768]	*a2 b2* crg1:*uded1*::*cbx^R^* [*pCM768*]	This study
521 crg1:*uded1* [pCM768]	*a1 b1* crg1:*uded1*::*cbx^R^* [*pCM768*]	This study
518 crg1:*uded1* [pCMas-ssm1]	*a2 b2* crg1:*uded1*::*cbx^R^* [*pCMas-ssm1*]	This study
521 crg1:*uded1* [pCMas-ssm1]	*a1 b1* crg1:*uded1*::*cbx^R^* [*pCMas-sm1*]	This study
*udbp3* Mutants		
518Δ*udbp3*	*a2 b2* Δ*udbp3*::*hph^R^*	This study
521Δ*udbp3*	*a1 b1* Δ*udbp3*::*cbx^R^*	This study

* *a1 b1* = mating type loci genotype, *a2 b2* = mating type loci genotype, *hph^R^* = hygromycin resistance, *cbx^R^* = carboxin resistance.

## Data Availability

Data is contained within the article and Appendix A.

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
