# Peer review of "Characterization of RNA Helicase Genes in *Ustilago maydis* Reveals Links to Stress Response and Teliospore Dormancy"

_ijms, 2025, doi:10.3390/ijms26062432_

Round 1

Reviewer 1 Report

Comments and Suggestions for Authors

The authors described the analysis of RNA Helicases in U. maydis. Previously they found their regulation related with teliosporte germination. The mutants displayed pleiotropic phenotype, and also important to mention the necessity to introduce a wild type copy of the gene under the regulation of crg1 promoter.

I found this paper interesting considering that there is not so much evidence about the function of this proteins in fungi in general. Besides U. maydis genome analysis revealed several related encoding RNA helicases genes.

I recommend to explorte the use another promoter like otef instead of using crg1 promoter. Considering that there is not much information about these genes in fungal pathongens. Or aa you mention try to supply L-arabinose during the initial steps of the pathogenicty test. 

Minor corrections to be done.

Line 322, S. pombe should be italicized

Figure 5, change the scale bar to a black colour. 

Line 429, Hordeum vulgare instead of Hordeum vulgara

Author Response

"I recommend to explorte the use another promoter like otef instead of using crg1 promoter. Considering that there is not much information about these genes in fungal pathongens. Or aa you mention try to supply L-arabinose during the initial steps of the pathogenicty test."

We thank the reviewer for these suggestions. An issue with the OTEF promoter is that it is constitutive, and what we required for uded1 was the ability to control the promoter, that is, to turn it on and off to assess whether the gene product is required for normal growth. We do acknowledge that a future experiment testing over-expression of a helicase could utilise OTEF,

Regarding the minor edits:

  1. Line 322, S. pombe should be italicized

We have italicized S. pombe.

  1. Figure 5, change the scale bar to a black colour

We have changed all scale bars in Figure 5 to black

  1. Line 429, Hordeum vulgare instead of Hordeum vulgara

We have corrected the spelling of Hordeum valgare

Reviewer 2 Report

Comments and Suggestions for Authors

In this body of work, the authors have characterized RNA helicases in Ustilago maydis and defined their role in osmotic stress response, and gene translation expression during dormancy and germination.

The work done is commendable, however here are some major comments :

The overall writing of this manuscript is very thick. A lot mentioned in the methodology section should be avoided in the result section to make it an easy read for the audience.

Why were two different WT strains were chosen to begin with? What is the rationale behind it?

The format of spot plate analysis in fig. 3, 4 and 6 is inconsistent. The authors have chosen to show both 518/521 in some and omitted from others.

All the observations need to end with a conclusion. For example; what do authors conclude from Line 202-207?

The authors switch back and forth between biological and technical replicates; lack of consistency.

Although the RT PCR and phenotypic-based assay findings support the data, the complete manuscript can be re-written more succinctly. The text for udbp3 mutant and used mutant characterization can be combined so that the redundancy around mating assay, and stress response is avoided.

Author Response

Comment 1: "The overall writing of this manuscript is very thick. A lot mentioned in the methodology section should be avoided in the result section to make it an easy read for the audience."

Response 1: We appreciate the perspective and suggestion of the reviewer and have made several changes in the results section to improve clarity. In a manuscript format that has the methods after the results we have been criticized previously for not including some details of the methods that allow the reader to understand what experiments were carried out so they can assess the data. As such, not all methods  mentioned were removed.

Comment 2 Why were two different WT strains were chosen to begin with? What is the rationale behind it?

We thank the reviewer for highlighting the lack of clarity in this section. We have edited the paragraph to improve clarity. Please see the changes in what is now results section 2.4. Note that we have highlighted the use of both wildtype strains elsewhere in the manuscript to emphasize the need in this fungus to have sexually compatible haploids (two different WT strains) to assess biological functions such as mating and pathogenesis.

Comment 3 "The format of spot plate analysis in fig. 3, 4 and 6 is inconsistent. The authors have chosen to show both 518/521 in some and omitted from others."

We thank the reviewer for pointing out this apparent inconsistency which also indicated that we could improve the clarity of explaining the data presented. To do this, we have made edits to the manuscript, including the addition of new supplementary figures to clarify and support the results presented. Below is a brief explanation of the figures

Figure 3 – This figure shows the impact of osmotic stress on both the 518 and 521 wildtype and deletion strains for udbp3 because we did see a difference between the two.

Figure 4A – In this experiment, we did not see a difference in growth between the 518 and 521 strains and opted to show a representative image. We have added text to reflect that we did not observe a difference in growth. We have also created Figure S6 to show the impact of growth in repressive growth conditions for both 518 and 521 deletion strains.

Text was also added to indicate that we observed the same growth phenotype in the 521 uded1 mutants as the 518 uded1 mutants. We now include Figure S7 to show the same growth assay as Figure 5 but with the 521 uded1 mutants. We also indicate in the Figure 5 caption that the growth assay for the 521 uded1 mutants can be found in Figure S7.

Figure 6 spotting format – The format of this assay is different from Figures 3 and 4 because the experiment is different. Here, a mating assay was performed with the uded1 mutants. The 518 and 521 strains are sexually compatible and when they are mixed, they fuse to create a dikaryon. Successful fusion results in a white fuzzy spot. The column of spots indicates the 518 strains, and the row of spots are the 521 strains. The intersection of the column and row is the spot where the two strains are mixed.

Comment 4: "All the observations need to end with a conclusion. For example; what do authors conclude from Line 202-207?"

Response 4: We agree with the reviewer and have added conclusion statements where missing.  Please note that the line numbers have changed with the edits and tracking changes in the included edited document.  As an example of the changes  we include the edited version of the paragraph indicated by the reviewer here:

Ded1 in S. cerevisiae is an essential protein for cell viability. To assess the importance of uded1 in U. maydis, we attempted to create uded1 deletion mutants using the Kämper [35] homologous recombination-based method. These attempts resulted in no viable mutants. A two-step gene deletion method described by Ostrowski and Saville [14] was used to create uded1 deletion mutants. This method involved first creating expression mutants where the ectopic expression of uded1 was placed under the control of a carbon-sensitive inducible promoter and integrated at the ip locus. These expression mutants (crg1:uded1) were created in the sexually compatible 518 and 521 wildtype strains so that we could later assess the impact of gene alterations on the ability of the fungus to mate and infect Z. mays. Homologous recombination was then used to replace the native uded1 with a hygromycin B resistance cassette in the crg1:uded1 strains. Growth of deletion strains (Δuded1 crg1:uded1) was observed when incubated in the presence of L-arabinose (permissive growth conditions). In repressive growth conditions, where D-glucose is the carbon source, these mutants were effectively deletion mutants (Figure 4A). Under repressive growth conditions, the deletion mutants showed reduced growth at the 10-1 and 10-2 dilutions compared to the wildtype and expression strains. No difference in growth was found between the 518Δuded1 crg1:uded1 and 521Δuded1 crg1:uded1 strains (Figure S6). The ability to control when the ectopically integrated uded1 is expressed thus enabled us to determine that uded1 is required for full cell viability and growth in U. maydis.

The authors switch back and forth between biological and technical replicates; lack of consistency.

We thank the author for pointing this out. There are instances where we interpret replicates as technical that others consider biological for example, repeating the experiment using the same strains we interpret as a technical replicate even though any variation in environment would change the biological response making this easily interpretable as a biological replicate.

We have clarified in the figure captions for Figures 3, 4, 5,6,  that technical replicates of the assay were conducted, and we are presenting the representative of those technical replicates.

Comment 6: "Although the RT PCR and phenotypic-based assay findings support the data, the complete manuscript can be re-written more succinctly. The text for udbp3 mutant and used mutant characterization can be combined so that the redundancy around mating assay, and stress response is avoided."

We see the point that the reviewer makes. 

We have combined the Results sections 2.2 and 2.3 and renamed Results 2.2 to “udbp3 characterization”. and we have renamed Discussion section 3.1 to “The role of udbp3 in osmotic stress response” and edited it a bit to improve clarity and make it somewhat more succinct.
